# The Senior Companion Program Plus (SCP Plus): Examining the Preliminary Effectiveness of a Lay Provider Program to Support African American Alzheimer’s Disease and Related Dementias (ADRD) Caregivers

**DOI:** 10.3390/ijerph20075380

**Published:** 2023-04-03

**Authors:** Ling Xu, Noelle L. Fields, Ishan C. Williams, Joseph E. Gaugler, Alan Kunz-Lomelin, Daisha J. Cipher, Gretchen Feinhals

**Affiliations:** 1School of Social Work, University of Texas at Arlington, Arlington, TX 76019, USA; 2School of Nursing, University of Virginia, Charlottesville, VA 22903, USA; 3School of Public Health, University of Minnesota, Minneapolis, MN 55455, USA; 4College of Nursing and Health Innovation, University of Texas at Arlington, Arlington, TX 76019, USA; 5The Senior Source, Inc., Dallas, TX 75219, USA

**Keywords:** ADRD, caregiving, lay provider, intervention, culturally congruent

## Abstract

Objectives: A culturally informed, peer-led, lay provider model, the Senior Companion Program (SCP) Plus, was implemented to decrease caregiving burden/stress and improve coping skills and social support for African American ADRD caregivers. This study reported the preliminary effectiveness of this intervention. Methods: An explanatory sequential mixed methods design was used in this study, and a randomized control trial was conducted for the SCP Plus intervention among participants in three sites (*n* = 20). A subsample of participants (*n* = 7) consented to a qualitative interview about their experiences with the intervention. Wilcoxon signed-rank tests, Friedman tests, and one-way repeated measures ANOVA were computed for quantitative analyses. Thematic analysis was used for the qualitative interviews. Results: Results demonstrated that knowledge of AD/dementia (KAD) and preparedness for caregiving were significantly improved for all senior companions in the intervention group. Results also showed that caregivers in the intervention group reported significantly decreased caregiving burden, as well as increased KAD, satisfaction with social support, and positive aspects of caregiving. Themes from the qualitative interviews included: learning new skills about caregiving, gaining knowledge about ADRD, and benefits for the dyad. Discussions: Findings from this study implied that SCP Plus was a promising model for African American family caregivers as it benefits both the SC volunteers and the African American ADRD family caregivers.

## 1. Introduction

Alzheimer’s disease and related dementia (ADRD) is one of the most challenging chronic health conditions in the United States, and its prevalence for Americans age 65 and older is projected to grow from 58 million in 2021 to 88 million by 2050 [1]. Considerable disparities exist in the diagnosis and prevalence of ADRD in older African American populations compared to older White populations. These disparities extend to family caregivers of persons with ADRD. Family caregivers face not only the prospect of needing to provide assistance with activities of daily living, but also with the challenges of coping with the profound cognitive and behavioral changes of the care recipients (CRs) with ADRD. Additionally, health and socioeconomic disparities, as well as structural racism, influence access to quality health care, and other resources for African Americans [2,3].

There are many empirically supported psychosocial and psychoeducational interventions for ADRD family caregivers [4,5,6]. However, most published evidence-based interventions do not include racially diverse samples [7], so it is unclear how these interventions specifically impact African American caregivers [8]. It is important that interventions for African American caregivers are centered around the needs of their CRs as well as finding community support to assist with navigating resources and information related to dementia. Brewster et al. [9,10] call for future research to evaluate and help develop culturally tailored interventions that can help reduce the depression, anxiety, and burden of African American caregivers and their families. Culturally congruent interventions may help earn this trust and improve the well-being of African American caregivers and their families [11]. Given the need for caregiver interventions for family caregivers that address the cultural aspects of caregiving, peer-led, lay provider models may offer promise.

Peer-based interventions can reduce caregiver isolation and provide a space for caregivers to exchange dementia-related knowledge [12]. There is a need to better understand the effectiveness of volunteer-led programs, specifically the impact they can have on African American informal caregivers. One well-established peer-based volunteer program is the Senior Companion Program (SCP) (now known as the AmeriCorps Seniors Senior Companion Program). The SCP is funded by the Corporation for National and Community Service (CNCS) (now operating as AmeriCorps since 2020) and is one of three National Senior Service Corps programs (now called AmeriCorps Seniors programs) in the United States. The SCP assigns adults aged 55 and older who are lower income to serve in the homes of frail older adults as volunteer companions [13]. Senior companions’ tasks include providing respite to family caregivers, assisting with grooming, eating, exercise, and offering peer support [13]. As volunteers, senior companions were offered a non-taxable stipend of USD 2.65 per hour at the time of our study (will change to USD 4.00/hour on 1 April 2023) (AmeriCorps, n.d.) [14].

Although the SCP serves persons with memory impairment, including ADRD, there is no specific program component related to ADRD family caregiving in terms of education and training [15]. The SCP also serves older adults and their families from diverse communities [16]. Thus, the SCP offers an opportunity to augment existing senior companion services and supports to include an intervention for African American ADRD caregivers. To address the need for accessible and culturally-congruent ADRD caregiver interventions, we developed the Senior Companion Program Plus or SCP Plus, a peer-led psychoeducational model provided by senior companions. In SCP Plus, senior companions were first provided with two days of training that included nine modules of content (see Table 1 and description in the Methods part) to increase their knowledge of ADRD and skills for ADRD caregiving as well as to prepare them for the logistics of the intervention. The trained senior companions then delivered what they had learned from the training to the African American ADRD caregivers. The contents of SCP Plus were based on a culturally-informed ADRD caregiver program for African American caregivers [17] and adapted by Fields and colleagues for a pilot study using volunteers from the SCP [18,19]. Central to the design and implementation of the SCP Plus is the concept of leveraging volunteers (i.e., senior companions) to support ADRD family caregivers; the program takes into account that some African American ADRD caregivers provide care with fewer financial resources and may face challenges related to accessing and affording other resources [20]. Using an explanatory sequential mixed methods design and randomized control trial of SCP Plus intervention, this study reported on the preliminary effectiveness of SCP Plus for senior companions and family caregivers. Specifically, the present study aimed to:Examine whether the SCP Plus helped senior companions gain more knowledge of dementia, preparedness of caregiving and sense of competence. We hypothesized that senior companions will improve significantly on these three outcomes.Investigate whether the SCP Plus helped African American family caregivers after the intervention when compared to the service as usual (SAU) control group on the major study outcomes. We hypothesized that the SCP Plus will significantly reduce caregiving burden/stress and appraisal of problem behaviors, increase knowledge of ADRD, as well as improve coping skills, level of social support, positive aspects of caregiving, cultural justifications for caregiving, and overall well-being in caregivers.Explore the experiences of family caregivers and senior companions with the SCP Plus after intervention.

## 2. Methods

### 2.1. Study Design

The revised sociocultural stress and coping model [21,22] was used to guide the present study. This theoretical framework focuses on generic stressors that often stem from challenging ADRD-related behaviors that many caregivers experience as well as cultural coping styles and social supports. Figure 1 illustrates the “targets” of the connecting activities of the present intervention as specified within the conceptual framework (e.g., coping style, social support, and well-being). For more information on how this conceptual model was applied in the present study, please refer to our prior studies [18,19].

An explanatory sequential mixed methods design was used in this study. Quantitative data were collected at baseline prior to the intervention, at the end of the intervention, and at 6 months post intervention. To further explore the experiences of family caregivers and senior companions with the SCP Plus after the intervention, we conducted follow-up qualitative interviews with participants from the intervention who were willing to participate (both caregivers and senior companions; *n* = 7). The qualitative interviews were conducted through telephone calls, which lasted approximately 60–90 min.

Though previous pilot study showed SCP Plus was promising, it did not use randomized controlled trial and thus could not tell if the SCP Plus intervention was effective in helping ADRD caregivers with burden/stress and coping skills. The present study thus expanded the program to three states to conduct a randomized evaluation of efficacy to determine the effects of SCP Plus on African American ADRD family caregivers’ stress and burden, coping skills, and social support. After the SCP directors at each site identified the senior companion–caregiver dyads who met inclusion criteria, participants were randomly selected. The project coordinator and project assistant then contacted each dyad. Participant dyads were offered the opportunity to participate in the study, guided through the informed consent process, administered a baseline survey via phone, and randomized to one of two groups. The first group of senior companion–caregiver dyads received the intervention, which included service as usual (SAU) plus the psychoeducational program for ADRD caregivers (SCP Plus), whereas the second group served as the control group, which included those who only received SAU as part of the SCP. A randomization list was generated, and group assignments were put in sealed envelopes from which the project coordinator drew senior companion–caregiver dyads who were randomly assigned to either SCP Plus or SAU.

The design for this study is depicted below in scientific notation. Data were collected by phone at baseline prior to the intervention (O_1_), at the end of the intervention (O_2_), and 6 months following the intervention (O_3_).
G_1_ O_1_ SCP Plus O_2_ O_3_
G_2_ O_1_ Control O_2_ O_3_

### 2.2. Intervention

The SCP Plus included a 12 h, in-person training with the senior companions only in the intervention group. In the informed consent form approved by the Institutional Review Board (IRB), the senior companion–participants were notified that they would be randomly assigned to either control group or intervention group. By signing the informed consent, they confirmed their understanding that they would receive the training only if they were assigned to the intervention group.

The training was conducted in a group format over the course of two days at each participating site. Three trainings were conducted (Dallas, TX, USA—May 2019; Little Rock, AR, USA—September 2019; Baton Rouge, LA, USA—January 2020). Two researchers lead the training, which features several types of teaching strategies, including rapport building through informal conversations at breakfast and lunch, the use of a written manual, time for questions, and both didactic and hands-on participation (e.g., role play). Nine topics are covered over the course of the two-day training [18,19]. After the completion of the SCP Plus training, each senior companion delivered the in-home psychoeducational sessions that covered nine topics (see Table 1 for the module topics). Senior companions met for one hour each week with their paired caregivers and only one topic per week was covered over a three-month period.

**Table 1 ijerph-20-05380-t001:** The Contents of Senior Companion Plus (SCP Plus) (adapted from Morano & King, 2010 [17]).

Modules
1. Myths and Facts about Alzheimer’s Disease
2. Managing Home Care and Home Safety
3. Management of Problematic Behaviors
4. Communicating with Healthcare Professionals
5. Community-based Services and Supports
6. Communication Skills with Persons with ADRD and Family Members
7. Coping Skills for the Emotional Consequences Associated with Caregiving
8. Coping with Expectation and Finding Meaning and Purpose in Caregiving
9. Summary of Modules 1–8

The SCP Plus intervention also include a weekly, 60-min psychoeducational module that is delivered face-to-face to ADRD caregivers by senior companions across 9 weeks and focuses on education about ADRD, behavior management, communication skills, and aspects of providing care that enhance meaning such as spirituality. Following completion of these sessions, we anticipated that participants would acquire knowledge and skills related to ADRD caregiving through psychoeducational modules that address the myths and facts about dementia, management of problematic behaviors of persons with dementia, development of communication skills with health professionals, information about community-based services, and positive coping strategies such as the use of spirituality. By acquiring knowledge and skills, the intervention was expected to positively influence the coping style and social support of the participants leading to improvements in ADRD caregiver well-being.

### 2.3. Sample

The study took place in collaboration with the SCPs in Dallas, Texas, Little Rock, Arkansas, and Baton Rouge, Louisiana. Study participants were senior companion–caregiver dyads that included senior companions recruited from SCPs in three sites as well as African American family caregivers of community-dwelling older adults with ADRD who were part of an SCP. The final sample size for the present study consisted of 20 dyads at baseline. Baseline data were collected and the intervention began in Dallas in May 2019, in Little Rock in September 2019, and in Baton Rouge, in January 2020. However, due to the challenges and risks of conducting an in-person intervention during COVID-19, data collection was suspended in March 2020. Thus, only 11 caregivers completed the post- and follow-up tests. Additionally, the Baton Rouge site intervention was not completed due to COVID-19 (e.g., senior companions were not allowed to meet with families due to health risks, the university did not permit the researchers to travel, the IRB did not permit in-person data collection). Of this sample, a total of 7 caregivers agreed to and completed a follow-up qualitative interview. Appendix A reports the descriptive information of senior companions in the three sites (*n* = 20), and Appendix A reports the descriptive information of caregivers in the three sites (*n* = 20) (see Appendix A).

### 2.4. Measurements

All caregiver outcomes were collected in pre-, post-, and 6-month following-up tests. For senior companions, the outcomes were measured in pre- and post-tests after attending two days of training. Standardized measures were used to assess these outcome variables and summed scores were calculated for final analyses.

#### 2.4.1. Outcomes Measures for Senior Companions

*Knowledge of Dementia* was measured by *Knowledge of AD/dementia scale* (KAD) to assess attitudes and beliefs regarding AD/dementia [23]. It consists of 14 statements that were rated by participants as false or true. Results have shown good reliability and validity of this scale in assessing attitudes and beliefs regarding AD/dementia among different ethnic minority caregivers [24].

*Sense of competence* was measured by the subscale of “satisfaction with one’s own performance as a caregiver from The Sense of Competence Questionnaire (SCQ) [25]. SCQ has been developed and repeatedly used to measure the competence in caregiving of dementia patients, and it has proved to be reliable and valid for this population [26,27]. It also shows a high degree of correspondence with classifications made by a panel of 39 experts, including professional caregivers and clinical researchers [27].

*Preparedness for caregiving* was measured by The Preparedness for Caregiving Scale (CPS) [28], which is a caregiver self-rated instrument that consists of eight items that asks senior companions how well prepared they believe they are for multiple domains of caregiving. Responses are rated on a 5-point scale with scores ranging from 1 (not at all prepared) to 5 (very well prepared). This scale has demonstrated moderate to high reliability [29,30] and validity [28].

#### 2.4.2. Outcomes Measures for Caregivers

*Dementia caregiver burden and/or stress* was measured by The Zarit Burden Interview (ZBI) [31]. The ZBI was developed to measure burden among caregivers of community-dwelling persons with dementia [32]. The 22-item version of ZBI was used most often in the literature and is a global measure of burden. The items are scored from 0 to 4, with higher scores indicating higher levels of distress. Total scores ranged from 0 to 88. The ZBI has shown good validity and reliability among various minority dementia caregivers [33] but is especially valid for African American caregivers [34].

*Coping skills* were measured by the Brief Cope subscales. The Brief Cope questionnaire consists of 28 items measuring the ways/strategies caregivers have been coping with the stress in their life with 4-point scale ranging from “I have not been doing this at all” (1) to “I have been doing this a lot” (4). Studies show good validity and reliability using the Brief Cope in caregivers of persons with dementia [35].

*Social support* was measured with 13 items from four domains: received support (Barrera et al., 1981 [36]), satisfaction with support [37,38], social support network [39], and negative interactions [37]. Social support network was measured with categories for the number of relatives or friends that the caregiver interacts with during the month on a 6-point scale (none, one, two, three or four, five to eight, nine or more). All other items had a 4-point scale that ranges from 0 (never) to 3 (very often) [40]. Results have shown these social support measures to be valid and reliable among African American caregivers [41].

*Caregiver appraisal* consisted of two components: appraisal of problem behaviors and appraisal of benefits. Regarding appraisal of problem behaviors, caregivers were asked whether problems had occurred during the past week (yes or no) measured by a modified version of the Revised Memory and Behavior problem Checklist (RMBPC) [42]. If caregivers responded yes, they then were asked to rate how much the problem “bothered or upset” them on a 0 (not at all) to 4 (extremely) scale. Results suggest that the RMBPC is a valid and reliable measure for different ethnic minority caregivers [43]. It has shown to be especially stable with African American caregivers [44]. Appraisals of benefits from caregiving was assessed with the Positive Aspects of Caregiving (PAC) scale [45]. The PAC consists of 9 items phrased as statements about the caregiver’s mental-affective state in relation to the caregiving experience. Each item begins with the stem “Providing help to the care recipient has…” followed with specific items (e.g., made me feel useful, enabled me to appreciate life more). Each item is rated on a 1–5-point Likert scale ranging from “disagree a lot” (1) to “agree a lot” (5). Results show that the PAC is a valid and reliable measure for different ethnic minority ADRD caregivers [33], and especially useful for African American ADRD caregivers [46].

*Cultural justifications for caregiving* was measured by the Cultural Justifications for Caregiving Scale (CJCS) [47], which is a 10-item measure designed to assess caregivers’ cultural reasons and expectations in providing care. Each item is measured using a 1–4-point Likert scale ranging from “(1) strongly disagree” to “(4) strongly agree.” Summed scores were used, which range from 10 to 40, with higher scores indicating stronger cultural reasons for giving care. Results have shown a good reliability and validity score of this scale with African American caregivers [47].

*Caregiver well-being* was measured by Caregiver Well-being Scale (CWBS). The 16-item short-form CWBS [48] was developed from the original 43-item CWBS [49] to help family caregivers, clinicians, and researchers to identify areas of caregiver strength and areas in which additional support is needed. Studies show that the short version CWBS has good content and construct validity as well as reliability, regardless of the ethnicity of the caregivers [48]. All of the 16 items were measured by a 5-point scale ranging from “rarely” (1) to “usually” (5).

*Background characteristics* of both caregivers and SCs were also collected (see Appendix A). For caregivers, information about the sociodemographic characteristics of caregiver and CRs, length of caregiving relationship, number of hours spent in caregiving, physical health, etc. were collected in the baseline surveys. The Dementia Severity Rating Scale (DSRS) with 12 items was also used to measure functional abilities of the CRs. Similarly, sociodemographic information of senior companions was also collected, including daily working hours, years of being senior companion, physical health, and help with CRs’ activities of daily living (ADLs) and instrumental activities of daily living (IADLs).

### 2.5. Data Analyses

An *a priori* power analysis using G*Power 3.1.9 indicated that a total sample size of 114 participants would be required based on an anticipated small effect size of *f* = 0.12, alpha = 0.05, and beta = 0.20, with two groups and three repeated measures. Post-hoc power analyses indicated that the actual study effect sizes for the primary outcomes were larger than anticipated overall, with effect sizes ranging from *d_z_* = 0.05 to 0.69.

For quantitative survey data, univariate descriptive analyses of demographic information were first conducted to better understand the senior companion and caregiver samples. Because of the small sample size, medians were reported instead of means and standard deviations. We then computed Wilcoxon signed-rank tests to discover if the main outcome variables for senior companion participants changed after the training between the control and SCP Plus groups. These analyses were repeated for the caregiver participants in the control and SCP plus groups. Friedman tests were computed for non-normally distributed outcome variables and one-way repeated measures ANOVA were computed for normally distributed outcome variables. Post-hoc between-group comparison tests were also conducted to test changes between pre- and post-tests, post- and follow-up tests, as well as pre- and follow-up tests. The study alpha was set to 0.05.

For qualitative interview data, two members of the research team followed Braun and Clarke’s six-steps of thematic analysis [50] to explore the interview data. First, the two researchers independently read the transcripts to become familiar with the data. Second, they independently coded each interview and then met to review the initial codes until consensus was reached. Next, the researchers independently searched for themes (step three) and then met together to review (step four) and define the themes (step five) until consensus was reached. Step six involved summarizing and reporting the themes.

## 3. Results

### 3.1. Quantitative Results

Table 2 shows the key measurements in the pre- and post-tests for senior companions. Knowledge of AD/dementia (KAD) (*z* = −2.97, *p* = 0.003) and preparedness of caregiving (*z* = −2.81, *p* = 0.005) were significantly improved for senior companion participants in the intervention group (see Table 2).

Table 3 reports the key measurements for CGs in pre-, post-, and follow-up tests in Texas and Arkansas. Baton Rouge did not complete the intervention due to the SCP programming being suspended during the COVID-19 pandemic. As seen in Table 3, caregiver burden decreased over time (*F*(2,10) = 20.47, *p <* 0.001) especially between pre- and post- tests as well as between pre- and follow-up surveys. KAD (*χ*^2^(2) = 8.72, *p* = 0.01), satisfaction with social support (*χ*^2^(2) = 6.33, *p* = 0.04), coping skills *F*(2,12) = 4.10, *p* = 0.04), and PAC (χ^2^(2) = 11.39, *p* = 0.006) also increased significantly after the intervention. CJC were marginally significantly improved after SCP Plus intervention.

Statistical comparisons were also conducted among key measurements of caregivers between the control group and the SCP plus group in Texas and Arkansas. Only significant results for major outcomes are reported in Table 4. As shown in Table 4, four key measurements showed significant changes for caregivers in the SCP Plus group. Specifically, caregivers in the SCP Plus group reported a significant decrease in caregiving burden (*F*(2,6) = 17.65, *p* = 0.003), especially between pre- and follow-up tests, as well as increase in KAD (*χ*^2^(2) = 6.53, *p* = 0.04), satisfaction with social support (*χ*^2^(2) = 6.53, *p* = 0.04), and PAC (*χ*^2^(2) = 6.53, *p* = 0.04), especially between pre- and post-tests as well as between pre- and follow-up tests. Coping skills of caregivers differed after the program, a finding that approached significance (*F*(2,6) = 4.44, *p* = 0.07). Caregivers in the control group did not reveal any significant changes on any of the key measurements.

### 3.2. Qualitative Results

The qualitative analysis led to the emergence of three themes, which contextualize and further support the abovementioned quantitative results: (1) learning new skills about caregiving; (2) gaining knowledge about ADRD; and (3) benefits for the dyad. These three themes are expanded upon below with representative quotes.

The first theme was related to caregivers learning new skills as part of the SCP Plus. Skills included CGs’ changing their approach to supporting activities of daily living. For example, one caregiver shared:

“Well…[name of a senior companion] had a good suggestion for me that was perfect. …because Mom would have all these clothes all in her room and she was still pretty much dressing herself and [name of a senior companions] told me to start to kind of remove some of those clothes, but to start putting her outfits together, pants and shirt, you know, and then, you know, put it all together and that’ll make it easier for her [care recipient] to continue to dress herself. And so, yes, I learned that from her. I learned too some little things, especially when she was moving towards her dementia things… I say they like to ‘fiddle with things.’ So I learned some little traits to have her to do to occupy her fingers and that would would settle her down. So I learned some of those things as well.”(Caregiver, Dallas)

Similarly, another caregiver shared:

“It turned around a lot of things that I was doing that I found out I was doing incorrectly as far as taking care of Mom. For instance. Uh, trying to make her remember things when she couldn’t, and I learned in the I learned in the program that that was not the thing to do. It was more or less to allow her to remember it the way that she remembered it and not try to force her to remember something that she couldn’t. And so. But I yeah, that was just one of the things that stood out with me. So there were several things that that I that it helped me with that I changed in taking care of her that helped out a whole lot. I mean a whole lot it. It was a a lifesaver. The program was a lifesaver.”(Caregiver, Dallas)

Additionally, one caregiver reported changing their approach to handling challenging behaviors of the person with ADRD:

“A lot of things cleared up and a lot of things learned. How to handle a person with dementia, Alzheimer’s. You know, sometimes it takes for you to walk through their delusions for them to get to a better place, like even with that.”(Caregiver, Baton Rouge)

A second theme was related to education, particularly related to knowledge about ADRD. For example, one caregiver shared:

“Ohh man ohh it to be honest with you it was very eye opening about a lot of misconceptions about dementia and and honestly not just the misconceptions, but just some things that’s just not known., I mean it, it was so much in that that you learn about. Dementia. Alzheimer’s that you either ever knew. If it wasn’t that you never knew the the myth and the misconception of it”(Caregiver, Baton Rouge)

Another caregiver shared that they learned about other symptoms of ADRD. She shared, “You know, we we’re talking about Sundowning. And stuff like that. At night, you know, she [care recipient] was changed completely, completely changed. And I said yeah, that that that happens with me. I told her [Senior Companion] since she completely changes at night.” (Caregiver, Baton Rouge)

The final theme was related to the caregivers’’ perspective that the SCP Plus had benefits for the dyad. For example, one caregiver shared:

“I do feel like when we did that [SCP Plus], that it helped us both…I want to say it helped us better bond, to talk, to deal with grandmother…It helped us both to be able to talk to somebody else…that made a difference…It made a difference for her and I to be on the same page at the same time talking about Grandmother.”(Caregiver, Little Rock)

Similarly, a caregiver shared that “she (the Senior Companion) learned very much from the training”. (Caregiver, Baton Rouge)

## 4. Discussion

Guided by the sociocultural stress and coping model, this study examined the preliminary evidence of the benefit from the SCP Plus program that aimed at reducing caregiving burden of African American family caregivers who cared for a person with ADRD. Results indicated that SCP Plus is a promising program as it helped the senior companions gain knowledge of dementia and improved their preparedness for caregiving. SCP Plus also appeared to be associated with increases in caregiver participants’ knowledge of dementia, decreasing their caregiving burden/stress, as well as increasing satisfaction with social support and positive aspects of caregiving.

For volunteer lay providers, findings from the training component of the SCP Plus offer promise as the senior companions were able to gain knowledge and skills as part of the program. The training focused on improving the senior companions’ knowledge of dementia, sense of competence, and preparedness for caregiving. The qualitative interviews also underscored that caregivers perceived that senior companions benefitted from SCP Plus. Similar to findings from a dementia-focused adaption of a Minnesota-based SCP [51], our training results suggest that the SCP Plus training positively influenced the senior companions’ knowledge and preparedness for caregiving. The results are also consistent with other research that trained volunteers may offer promise in providing interventions for persons with ADRD and their family caregivers [12,52,53].

Both the quantitative and qualitative findings suggest that caregivers reported increased knowledge of dementia after participating SCP Plus. These findings are promising, as research suggests that there is a continued need for interventions that promote knowledge of ADRD among racial and ethnic minority groups [54]. Previous research also points toward the need for, and benefits of, improving knowledge of ADRD among ethnically diverse ADRD caregivers [55]. Moreover, barriers to education may be related to disparities in healthcare [56], as well as due to the cultural stigma surrounding ADRD [57], suggesting that culturally informed educational strategies are important.

Findings also suggested that the SCP Plus program significantly decreased their caregiving burden/stress. These results are consistent with a meta-analysis that reported a small significant effect of psychoeducational interventions on caregiver burden [58]. Furthermore, study findings corroborate a meta-analysis of non-pharmacological interventions for ADRD caregivers suggesting that psychoeducation was effective in reducing caregiver burden [59]. This study thus confirms that using volunteer-based lay providers to provide psychoeducational interventions to family caregivers had similar effects in reducing stress/burden compared to programs provided by professionals.

The study findings highlight the role of social support as part of the SCP Plus. One aspect of social support is the intentional linkage between the senior companions and the caregivers as part of the program. Senior companions are typically focused on providing care to the CR, versus the caregiver. For some participants, the lay provider, peer-led model may have provided the opportunity to build a stronger relationship between the caregiver and the senior companion. This is consistent with other research suggesting that peer support relationships have positive social impacts [60]. Additionally, the SCP Plus also encourages caregivers to reach out to other family members, friends, and community members for support through different communication strategies. Studies suggest that additional sources of support provided by a network of family members may enhance health outcomes among African American ADRD caregivers [61]. Moreover, many African American ADRD caregivers express a desire for family and social support [62,63].

Finally, the results from the SCP Plus related to the positive aspects of caregiving (PAC) are promising, as Cheng and colleagues [64] emphasize the importance of developing interventions that are “not solely focused on alleviation of distress but also on building strengths, good health, and a positive perspective on caregiving” (p. 70). The SCP Plus includes an emphasis on finding cultural satisfaction in caregiving, reflecting on the positive aspects of caregiving, and caregiving resilience. Finally, these results align with previous studies showing that psychoeducational and multicomponent interventions for ADRD caregivers have a broad range of significant effects [64].

The results of this study must be interpreted with caution, as there are several limitations that provide lessons for future studies. First, the final sample size was very small, due in part to the negative impact of COVID-19. Therefore, we could not control for any confounding variables when examining the preliminary effectiveness of the SCP Plus program. However, it should be noted that the post-hoc power analyses yielded higher than anticipated effect sizes, which yielded statistical significance in spite of the low sample size. Nevertheless, a larger sample is needed to better examine the effectiveness of the SCP Plus. Moreover, because the intervention and data collection were suspended due to COVID-19 in Louisiana, we could not compare the differences in the three sites as planned. The main difference of participants in the Louisiana site might be that its SCP is connected to large hospital systems compared to the other two completed sites, which are more a part of community agencies. This may potentially influence some of the outcomes for caregivers in Louisiana. Second, the sample included African American ADRD caregivers from three sites in one southern region of the U.S. It is thus limited to represent SCP in other States or regions. A larger and more representative sample would shed light on how the SCP Plus could be implemented in other regions. Third, only the post-test for senior companions were designed at the end of the training, which may limit the comparisons of the main outcomes for senior companions. Because there might be no or few changes in the 2 days before and after the training, we did not survey the senior companions in the control group. Future study can modify the design, as post-testing senior companions at the end of the program, as well as 6 months after the program ended, should be similar to the parts of caregivers. Though the focus of the project was caregivers, collecting more information on senior companions may help the Senior Corps in developing other ADRD-related services. Moreover, the measurement of knowledge of dementia can be updated to include more recent information on dementia. The Alzheimer’s Disease Knowledge Scale (ADKS) with 30 items is recommended to measure participants’ knowledge of ADRD for future studies. Lastly, telephone-based interventions can be considered in the future to accommodate the spatial and geographical limitations of face-to-face interventions and other restrictions (e.g., COVID-19).

## 5. Conclusions

The SCP Plus is one of the few studies to partner with the SCP to deliver an in-home ADRD psychoeducational intervention. Although other studies examined the use of senior companions to provide dementia-capable support to persons with ADRD and their families [51], the SCP Plus is one of the only community-based, culturally-congruent, lay provider psychoeducational interventions designed for African American ADRD caregivers. The preliminary effectiveness of this intervention amplifies the need to develop programs that are informed by a clear conceptualization of one’s culture. Being intentional about the potential sociocultural factors that can hinder or support caregivers who are disproportionately impacted by ADRD is critical to meet the needs of diverse caregivers [65]. Partnering with SCP for designing, testing, and evaluating ADRD programs and supports offer promise for providing potentially sustainable community-based interventions as there is an existing Senior Corps infrastructure across all 50 states in the U.S. The SCP Plus also bolsters support for the leveraging volunteers to help address service gaps for ADRD caregivers. Research suggests that few ADRD caregiving intervention studies have moved from the realm of research to practice settings [66] and that the use of organized and trained volunteers may offer a cost-effective, scalable approach to programs for persons with ADRD and their families [63].

## Figures and Tables

**Figure 1 ijerph-20-05380-f001:**
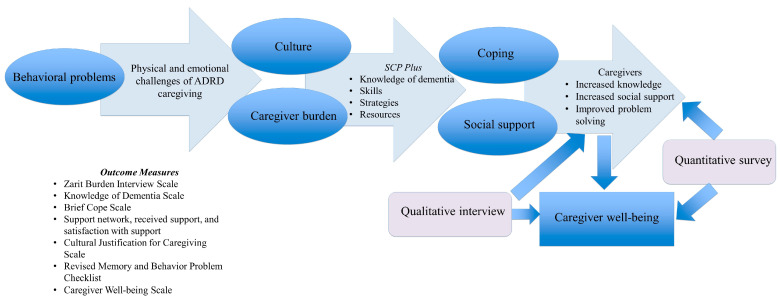
Theoretical Framework.

**Table 2 ijerph-20-05380-t002:** Key measurements in pre- and post-tests for SCs in control (*n* = 9) and SCP Plus (*n =* 11) groups.

Key Measurements	*Pre-Test*	*Post-Test*	Mean Differences (95% CI)	Statistical Test, *p*-Values *
	Medium (Range)	Medium (Range)		
Knowledge of AD/Dementia (Control)	8.00 (6–10)	N/A	N/A	N/A
Knowledge of AD/Dementia (SCP Plus)	10.00 (5–11)	12.00 (10–14)	2.82 (1.45–4.19)	*z* = −2.97, *p* = 0.003 **
Sense of Competence (Control)	46.0 (29–53)	N/A	N/A	N/A
Sense of Competence (SCP Plus)	46.00 (36–55)	45.00 (37–55)	0.64 (−0.90–2.18)	*z* = −0.67, *p* = 0.501
Preparedness for Caregiving (Control)	28.00 (23–32)	N/A	N/A	N/A
Preparedness for Caregiving (SCP Plus)	24.0 (21–39)	31.00 (23–40)	3.82 (1.72–5.92)	*z* = −2.81, *p* = 0.005 **

* The non-parametric test of Wilcoxon signed-rank tests were conducted. N/A: not applicable. ** *p* < 0.01.

**Table 3 ijerph-20-05380-t003:** Key measurements in pre, post, and follow-up tests for CGs in 2 sites (Texas and Arkansas).

Key Measurements	*Pre-Test (n = 11)*	*Post-Test (n =* *9)*	*Follow-Up Test (n =* *7)*	Statistical Test ^a^
	Median (Range)	Median (Range)	Median (Range)	
ADLs of CR	2.5 (0–6)	3.0 (0–6)	6.0 (1–6)	*χ*^2^(2) = 3.12, *p* = 0.21
IADLs of CR	8.0 (6–8)	8.0 (7–8)	8.0 (7–8)	χ^2^(2) = 2.00, *p* = 0.37
Knowledge of AD/Dementia (KAD)	9.0 (7–13)	12.0 (8–14)	11.0 (10–14)	χ^2^(2) = 8.72, *p* = 0.01 ^3,^*
Caregiving burden (Zarit burden)	39.0 (8–63)	22.0 (10–45)	13.5 (4–30)	*F*(2,10) = 20.47, *p* < 0.001 ^1,3,^***
Social support				
Social support network	14.0 (3–30)	19.0 (10–28)	21.0 (17–27)	*χ*^2^(2) = 0.96, *p* = 0.62
Received social support	5.0 (3–7)	5.0 (4–7)	4.0 (3–4)	χ^2^(2) =3.90, *p* = 0.14
Satisfaction with social support (SSS)	6.0 (0–12)	12.0 (3–12)	12.0 (5–12)	χ^2^(2) = 6.33, *p* = 0.04 ^1,3,^*
Negative interactions	3.0 (0–11)	1.0 (0–6)	0.0 (0–6)	χ^2^(2) = 3.36, *p* = 0.19
Coping skills	75.0 (55–98)	85.0 (74–101)	85.0 (61–101)	*F*(2,12) = 4.10, *p* = 0.04 ^1,3,^*
RMBPC	10.0 (7–13)	10.0 (1–18)	11.5 (3–17)	*F*(2,10) = 0.28, *p* = 0.76
RMBPC_bother	8.0 (0–36)	10.0 (0–32)	4.5 (0–40)	*F*(2,10) = 0.12, *p* = 0.89
Positive aspect of caregiving (PAC)	36.0 (23–49)	49.0 (30–55)	52.0 (36–55)	χ^2^(2) = 10.16, *p* = 0.006 ^1,3,^**
Cultural justifications for caregiving (CJC)	31.0 (21–37)	36.0 (13–40)	39.0 (26–40)	χ^2^(2) = 4.88, *p* = 0.09 £
Well-being of caregiver (WBC)	63.0 (31–76)	67.0 (42–80)	70.0 (43–76)	χ^2^(2) = 4.75, *p* = 0.09 £

^a^ Friedman tests were computed for non-normally distributed data. One-way repeated measures ANOVA were computed for normally distributed data. ^1^ Significant differences between pre- and post-tests. ^3^ Significant differences between pre- and follow-up tests. £ *p* < 0.10, * *p* < 0.05, ** *p* < 0.01, *** *p* < 0.001.

**Table 4 ijerph-20-05380-t004:** Key measurements that showed significant changes for CGs in 2 sites (Texas and Arkansas) between control group (*n* = 3) and SCP Plus group (*n* = 4) ^a,b^.

Key Measurements	*Pre-Test*	*Post-Test*	*Follow Up-Test*	Statistical Test ^a,b^
	Median	Median	Median	
KAD (Control)	9.0	10.5	12.0	χ^2^(2) = 3.20, *p* = 0.20
KAD (SCP Plus)	9.5	12.0	11.0	χ^2^(2) = 6.53, *p* = 0.04 ^1,^*
ZBI (Control)	18.0	18.5	7.0	F(2,2) = 2.94, *p* = 0.25
ZBI (SCP Plus)	40.0	32.0	15.0	F(2,6) = 17.65, *p* = 0.003 ^3,^**
Coping (Control)	73.0	82.0	90.0	F(2,4) = 0.42, *p* = 0.68
Coping (SCP Plus)	81.0	96.0	79.0	F(2,6) = 4.44, *p* = 0.07 £
SSS (Control)	9.0	12.0	12.0	χ^2^(2) = 0.67, *p* = 0.72
SSS (SCP Plus)	3.5	4.0	11.5	χ^2^(2) = 6.53, *p* = 0.04 ^1,3,^*
PAC (Control)	42.0	49.0	52.0	χ^2^(2) = 3.8, *p* = 0.15
PAC (SCP Plus)	33.5	49.0	52.5	χ^2^(2) = 6.53, *p* = 0.04 ^1,3,^*

^a^ Friedman tests were computed for non-normally distributed data. One-way repeated measures ANOVA were computed for normally distributed data. ^b^ Non-significant results on other outcomes were not reported in this table. ^1^ Significant differences between pre- and post-tests ^3^ Significant differences between pre- and follow-up tests. £ *p* < 0.10, * *p* < 0.05, ** *p* < 0.01.

## Data Availability

This research did not use a publicly available dataset.

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
