# Peer review of "The Senior Companion Program Plus (SCP Plus): Examining the Preliminary Effectiveness of a Lay Provider Program to Support African American Alzheimer’s Disease and Related Dementias (ADRD) Caregivers"

_ijerph, 2023, doi:10.3390/ijerph20075380_

Round 1

Reviewer 1 Report (Previous Reviewer 3)

Revisions are satisfactory

Reviewer 2 Report (Previous Reviewer 1)

The authors have addressed my previous concerns.

Reviewer 3 Report (Previous Reviewer 2)

The manuscript has been improved and warrants publication without

> further revisions

This manuscript is a resubmission of an earlier submission. The following is a list of the peer review reports and author responses from that submission.

Round 1

Reviewer 1 Report

This is an interesting study targeting SC peers and CGs across 3 very small samples in 3 different states. I liked the introduction and the theoretical rationale, but the extreme smallness of the sample combined with the multitude of statistical tests/measures, limiting findings largely to those which were statistically significant made for an unneccessarily complex read. For example, is the detail in Tables 2 and 3 necessary, and more importantly, relevant to the study's outcomes?

The authors would do well to limit the analyses to a finite set of measures and report all findings in tables, not just those that reflect group differences. The psychometrics of most measures are missing, and it is unclear whether those Ss in the control group indeed received SCP training, as would be required via IRB review. In addition, what is the effect of COVID-related attrition at posttest in LA? Likely substantial given the small samples to begin with. It was unclear what the authors were saying on pp 12-13 regarding the construction of the posttest for SCs/not surveying SCs in the control group. No mention is made of the lack of generalizations to male SCs or GCs as well as to non-African Americans, and the substantial age range of the CGs.

That this study is essentially a pilot study needs to drive the extensive results section, where more targeted measures whose findings are presented in a balanced manner, are used. I do feel the qualitative findings nicely complement the quantitative ones.

There is enough here to warrant further revision, but the authors need to reel in the extensive detail in Tables 2 and 3 as well as their results.

Author Response

Comments and Suggestions for Authors

  1. This is an interesting study targeting SC peers and CGs across 3 very small samples in 3 different states. I liked the introduction and the theoretical rationale, but the extreme smallness of the sample combined with the multitude of statistical tests/measures, limiting findings largely to those which were statistically significant made for an unneccessarily complex read. For example, is the detail in Tables 2 and 3 necessary, and more importantly, relevant to the study's outcomes?

Response: Thank you for this comment. We simplified Tables 2 and 3 to make sure the information provided are concise and necessary. We also put Tables 2 and 3 as supplementary and deleted the descriptive information of Tables 2 and 3 in the text.

  1. The authors would do well to limit the analyses to a finite set of measures and report all findings in tables, not just those that reflect group differences. The psychometrics of most measures are missing, and it is unclear whether those Ss in the control group indeed received SCP training, as would be required via IRB review. In addition, what is the effect of COVID-related attrition at posttest in LA? Likely substantial given the small samples to begin with. It was unclear what the authors were saying on pp 12-13 regarding the construction of the posttest for SCs/not surveying SCs in the control group. No mention is made of the lack of generalizations to male SCs or GCs as well as to non-African Americans, and the substantial age range of the CGs.

Response: The senior companions in the control group did not receive SCP training. In the IRB approved informed consent form, the senior companion participants were notified that they would be randomly assigned to either control group or intervention group. They knew from the consent form that they will not receive the training if they were assigned to the control group. We added this information in the revised manuscript (see yellow highlights on page 7).

Because of COVID, intervention was suspended in March 2020, which was about after two weeks of the intervention started in Louisiana (LA). Therefore, we cannot ascertain any COVID-related attrition in LA since the study was closed.

Since this study was designed specifically for African American who are disproportionately affected by dementia, this study has no intention to be generalized to non-African American. The age range and gender proportion of the present study align with the national profiles of dementia caregivers, therefore, no need of generalization to male caregivers or substantial age.

  1. That this study is essentially a pilot study needs to drive the extensive results section, where more targeted measures whose findings are presented in a balanced manner, are used. I do feel the qualitative findings nicely complement the quantitative ones.

Response: Thank you for this positive feedback. Because of the small sample size of the quantitative survey, we agreed with the reviewer that the qualitative finding is a nice complement.

  1. There is enough here to warrant further revision, but the authors need to reel in the extensive detail in Tables 2 and 3 as well as their results.

Response: Thank you for this comment. We deleted the detailed description of Tables 2 and 3 in the main text in the revised manuscript. See responses to comment #1 above.

Reviewer 2 Report

I think this paper was well written although a bit too long in length for such a small sample size.  

On line 180 when describing the clients ADLS the range of 0-7 was given but basic ADLS are 6 so where did that 7 come from?

On line 504 it states IRB of XX University - what is XX

On line 522 and 523 the AmeriCorps reference is not numbered and not clear what citation it belongs to.

Author Response

Comments and Suggestions for Authors

  1. I think this paper was well written although a bit too long in length for such a small sample size.  

Response: Thank you for this positive comment. We cut this paper to make it shorter and more concise in the revised manuscript. For example, we deleted description of the samples of senior companions and caregivers, and also simplified Tables 2 and 3.

  1. On line 180 when describing the clients ADLS the range of 0-7 was given but basic ADLS are 6 so where did that 7 come from?

Response: In this study, we used the ADLs measurement from Resources for Enhancing Alzheimer's Caregiver Health II (REACH II) study. In REACH II study, the ADL items are based on the list of activities originally used by Katz, Ford, Moskowitz, Jackson, & Jaffe (1963), but "dressing" is asked separately for lower and upper body. These 7 items are: help getting into or out of a bed, chair or wheelchair; help eating meals; help bathing; help dressing from the waist up; help dressing from waist down; and help toileting. Each item was measured by 7 items (0 = no, 1 = yes). Therefore, the total sum scores ranged from 0-7.

However, REACH II also suggest that these two items on dressing can be combined into one category – whether assistance is needed in dressing – in calculating the total ADL score. To be consistent with the literature, we combined these two dressing categories and calculated the sum scores as 0-6. See changes in Supplemental Table 1.

Katz, S., Ford, A. B., Moskowitz, R. W., Jackson, B. A., & Jaffe, M. W. (1963). Studies of illness in the aged: the index of ADL: a standardized measure of biological and psychosocial function. JAMA, 185(12), 914-919. doi:10.1001/jama.1963.03060120024016

  1. On line 504 it states IRB of XX University - what is XX

Response: We used XX for blind reviewing purpose. We added the institution name in the revised manuscript.

  1. On line 522 and 523 the AmeriCorps reference is not numbered and not clear what citation it belongs to.

Response: Thank you for this comment. We added this citation and reference.

AmeriCorps. (n.d.) Senior corps volunteers receive stipend increase. Retrieved from https://americorps.gov/newsroom/press-release/2020/senior-corps-volunteers-receive-stipend-increase

Reviewer 3 Report

This is a very interesting, small mixed methods study with potentially high impact findings if the program is scaled-up and deployed using the Senior Corps infrastructure. The manuscript would benefit from significant reorganization and editing with at least a sentence or two of detail about the intervention that is referenced but not described in the introduction.

As written (referring to title and abstract here) the reader has no idea that this is a mixed-method study until the middle of the methods section. The rationale for choosing a RCT design and more clearly saying what was randomized would also be helpful, as these details are easily missed in current version.

In Tables 2 & 3 it is not essential to see the characteristics split by site (that detail could be saved for an appendix at the end), and Tables 2 & 3 could be combined into a single table that compares the overall characteristics of the SCs and CGs. Given the loss of the largest study site (in Louisiana) due to COVID, the reader might be interested in knowing how different the Louisiana site was to the sites that completed the study, and this could be summarized in a few sentences.   

During editing, consider greatly reducing the use of acronyms in the abstract and throughout the paper, and writing more briefly. Examples include results section of the abstract, description of outcome measures, and ...

Figure 1 (Theoretical Framework) is difficult to read and might be updated with reference to the mixed methods study design and the various participant-stakeholders in the research (SCs CGs etc).

Alternately, given the limitations of small sample for quantitative analysis (control group, n=3 vs. intervention group, n=4), the main focus could be shifted to the qualitative results with brief description and discussion of quantitative results.

In summary, this is a very interesting project that I would like to read more about in the future, both as lessons learned from this pilot stage, and future larger scale research on this topic. 

Author Response

Comments and Suggestions for Authors

  1. This is a very interesting, small mixed methods study with potentially high impact findings if the program is scaled-up and deployed using the Senior Corps infrastructure. The manuscript would benefit from significant reorganization and editing with at least a sentence or two of detail about the intervention that is referenced but not described in the introduction.

Response: Thank you for this positive comment. A paragraph of intervention were briefly described in the introduction part on page 4-5 in the revised manuscript.

  1. As written (referring to title and abstract here) the reader has no idea that this is a mixed-method study until the middle of the methods section. The rationale for choosing a RCT design and more clearly saying what was randomized would also be helpful, as these details are easily missed in current version.

Response: We added mixed-method study design in the abstract as well as at the end of the introduction part (page 5). We also added the rationales for choosing a RCT design on page 6 in the revised manuscript.

  1. In Tables 2 & 3 it is not essential to see the characteristics split by site (that detail could be saved for an appendix at the end), and Tables 2 & 3 could be combined into a single table that compares the overall characteristics of the SCs and CGs. Given the loss of the largest study site (in Louisiana) due to COVID, the reader might be interested in knowing how different the Louisiana site was to the sites that completed the study, and this could be summarized in a few sentences.

Response: Thanks for the comments on Tables 2 and 3. We simplified these two tables, put them as supplemental material and deleted lengthy description of the tables in the revised manuscript. However, the two tables are hard to be combined because the characteristics of caregivers and senior companions are different.

The main difference of participants in Louisiana site might be that the Senior Companion Program is connected to large hospital systems compared to other two completed sites which are more part of community agencies. This may potentially influence some of the outcomes for caregivers in Louisiana. We added this in the Discussion part on pages 19-20.

  1. During editing, consider greatly reducing the use of acronyms in the abstract and throughout the paper, and writing more briefly. Examples include results section of the abstract, description of outcome measures, and ...

Response: Thank you for this suggestion. We revised the paper by reducing the use of acronyms (e.g., CGs and SCs).

  1. Figure 1 (Theoretical Framework) is difficult to read and might be updated with reference to the mixed methods study design and the various participant-stakeholders in the research (SCs CGs etc).

Response: We revised the theoretical framework to reflect the mixed method study design and caregiver participant (see revised figure 1).

  1. Alternately, given the limitations of small sample for quantitative analysis (control group, n=3 vs. intervention group, n=4), the main focus could be shifted to the qualitative results with brief description and discussion of quantitative results.

Response: We agreed with the reviewer. We revised the quantitative result part to make it more concise and briefer.

  1. In summary, this is a very interesting project that I would like to read more about in the future, both as lessons learned from this pilot stage, and future larger scale research on this topic.

Response: Thank you for this comment. We added a few lessons learnt from this project for future studies on pages 19-20 in the revised manuscript.
